# Direct Spread Spectrum Technology for Data Hiding in Audio [note 1]

**DOI:** 10.3390/s22093115

**Published:** 2022-04-19

**Authors:** Alexandr Kuznetsov, Alexander Onikiychuk, Olga Peshkova, Tomasz Gancarczyk, Kornel Warwas, Ruslana Ziubina

**Affiliations:** 1Department of Information Systems and Technologies Security, V.N. Karazin Kharkiv National University, Svobody Sq. 4, 61022 Kharkiv, Ukraine; kuznetsov@karazin.ua (A.K.); onik@karazin.ua (A.O.); o.g.peshkova@karazin.ua (O.P.); 2JSC “Institute of Information Technologies”, Bakulin St. 12, 61022 Kharkiv, Ukraine; 3Department of Computer Science and Automatics, University of Bielsko-Biala, Willowa St. 2, 43309 Bielsko-Biala, Poland; tgan@ath.bielsko.pl (T.G.); kwarwas@ath.bielsko.pl (K.W.)

**Keywords:** spread spectrum technology, data hiding, steganography, chip codes, audio

## Abstract

Direct spread spectrum technology is traditionally used in radio communication systems with multiple access, for example, in CDMA standards, in global satellite navigation systems, in Wi-Fi network wireless protocols, etc. It ensures high security and reliability of information transfer. In addition, spread spectrum technology provides the transmitted signals with a noise-like appearance, thus hiding the semantic content of the messages. We researched this technology for other implementations. The purpose of our study was to investigate new technologies for hiding data in multimedia files. In particular, we investigated the direct spread spectrum in the context of the development of methods for hiding data in audio containers. We considered various spreading sequences (chip codes) and also explored their use for hiding information in audio files. We conducted experimental studies and estimated the bit error rate (BER) in the recovered data. The article also evaluates the distortion of an audio container by the peak signal-to-noise ratio (PSNR). The results of our research enable us to find out which method of forming chip codes gives a lower BER with equal PSNR. We provide recommendations on the formation of spreading sequences to reliably and safely hide informational messages in audio files.

## 1. Introduction

Various techniques are used to conceal information [1,2,3]. The most common of them are cryptography techniques [4,5,6]. In this case, meaningful messages are converted into meaningless, noise-like forms, i.e., this becomes useless data for an unauthorized user (who does not know the secret cryptographic key). Thus, cryptography hides the semantic content of the transmitted data, whereas the very existence of messages is not hidden [3,7].

Another approach is to use steganographic techniques [3,8]. In this case, informational messages are hidden inside other data (also called containers, covers, carriers, etc.). Carriers can be presented, for example, in the form of various multimedia files (images, video, audio, texts, etc.) [9]. A range of applications exist with multimedia data, so the transfer of video or audio files on the internet does not cause any suspicion. At the same time, only the one who has the secret steganographic key knows that an informational message is hidden in these covers. It is similar to a suitcase with a double bottom; the second bottom is reliably disguised, and it is difficult to detect a hidden message without a secret key. Thus, steganography hides the fact that messages exist [2,8].

The present paper discusses steganographic information hiding techniques in which audio files act as multimedia containers. Hiding data in audio has great prospects. Firstly, such data are significantly popular on the internet. For example, we can use audio messages in messengers and social networks as covers. At the same time, the amount of hidden data can be quite large. In addition, there is a problem with copyright protection of audio content on the internet. Hiding digital watermarks (with a copyright label) in audio containers can be an effective solution. Thus, hiding data in audio is really very relevant and may have a commercial implementation.

In our experiments, we used the simplest audio signals recorded in the digital WAV format [10,11]. However, all our results and suggestions can be interpreted for a more general case with other audio container formats.

In order to conceal information, this work employs a special technology traditionally used in radio communication systems [12,13,14,15]. This is a direct spreading of the spectrum [16], which implements wideband modulation with pseudo-random sequences (chip codes) [17]. The correlation properties of chip codes guarantee the absence of mutual interference in the communication channel [18]. As a result, the radio communication system acquires numerous useful properties: noise immunity, high bandwidth and subscriber capacity, absence of mutual interference, environmental friendliness of communication, and many others [19].

Direct spread spectrum technology is implemented in many industry standards and applications, for example:Code division multiple access communication systems (IS-95, CDMA2000, WCDMA, DS-CDMA, TD-CDMA, TD-SCDMA, etc.);Global satellite navigation systems (American GPS, European Galileo, and Russian GLONASS);Family of wireless network protocols Wi-Fi (family of standards IEEE 802.11), and much more.

Direct spread spectrum technology can also be used in steganography [20,21,22]. In particular, in one of the first works [21] in this area, it was proposed to interpret the carrier as noise in the communication channel. In this case, the task of hiding information in multimedia files is equivalent to transmitting useful signals over communication channels with natural additive noise.

Currently, a large number of studies on steganography and the direct spreading of the spectrum have already been conducted. Various researchers studied this issue in relation to various computer applications. It turns out that the main characteristics of reliability and safety depend on the properties of the spreading sequences. In particular, in [23,24], we investigated various ways of generating chip codes in relation to cover images. In this paper, we continue these studies in relation to another type of multimedia data—audio carriers.

We studied the basic characteristics of an audio stegosystem for various spreading sequences. In our experiments, we estimate the bit error rate (BER) of hidden information data. This is the most important indicator of the quality of recovered messages. In addition, we evaluated the distortions of the audio container, which characterize the quality of multimedia data after hiding information messages in them. Therefore, we used the mean squared error (MSE) and the peak signal-to-noise ratio (PSNR). In this article, we demonstrated that the variety of ways of generating chip codes have different results for BER, MSE, and PSNR. It becomes an area for possible optimization, as the choice of the method of forming the spreading sequences is very important for various computer applications. Thus, the main goal of our work is to find the best chip codes that would minimize the BER for comparable PSNR.

## 2. Related Works

Hiding data in audio was investigated in many related papers, e.g., in papers [25,26,27,28,29], the simplest LSB techniques were studied, which, nevertheless, have some advantages. By encoding the LSB of the container, you can hide a very large amount of data. However, LSB techniques are also easily detected, and this is their main drawback. Various authors tried to improve LSB. In [26], the authors used augmentation operations to increase resistance to attacks. In [25], the LSB method was applied to the coefficients of the discrete wavelet transform. In [27], an LSB was proposed with an adaptive number of hiding bits for each audio sample (depending on the size of the hidden data, the size of the covering carrier, and the signal-to-noise ratio). In [28], binary data values were hidden in different places to increase security. In [29], the LSB technique was supplemented with data encryption. There are many other works in this area; however, each improvement of the LSB method leads, as a rule, to a decrease in the volume of hidden data [30,31].

Other approaches to hiding data in audio use various techniques: in [32], Cochlear Delay Characteristics were used to hide data; in [33], authors investigated bit reduction techniques; in [34,35,36], echo signals were used; in [37,38,39], data were hidden in the phases of audio signals. All these techniques have their advantages and disadvantages [31]. In particular, the amount of hidden data is significantly less than in LSB methods. However, the security of the hidden data is higher.

One of the prospective methods is steganography with direct spectrum expansion. These techniques combine high throughput and data security.

The first works [20,21,40] were devoted to the substantiation of the general concepts and structure of the steganosystem using a direct spread spectrum. Images were used to hide the messages. Already in those articles, the authors noted several conflicting requirements [41]. In particular, increasing the size of messages leads to increasing the corruption of containers; small BER values can be achieved only with a large “energy” of the embedded message, which significantly distorts the container. In order to eliminate some of these contradictions, the authors suggested using special filtering and correcting codes [41]. These and other ideas were developed in subsequent works. For example, in [42], correction codes were investigated; in [43,44], a discrete cosine transform (DCT) was implemented together with the direct spreading of the spectrum, etc. Papers [45,46] are devoted to the study of chaos. In works [47,48], hiding information in video data was studied, and in [42,49], audio signals were used for hiding information. Indeed, hiding messages in audio signals is a promising direction [50]; these methods can be used, for example, to protect the property rights of digital audio signals [51].

Note that in these numerous works, the properties of spreading sequences (chip codes) were almost not explored. The exceptions are our two recent works [23,24], in which we analyzed various methods of forming spreading sequences, as well as their influence on the characteristics of Spread Spectrum Image Steganography.

In this work, we continued our research and expanded their field to audio files. First, we looked at different ways to form spreading sequences and explored BER, MSE, and PSNR. Then, we explored possible compromises, such as how to minimize carrier distortion with low bit error rates. It turned out that the Walsh–Hadamard expansion sequences were best suited for this, and our experiments clearly demonstrated this.

## 3. Materials and Methods

Direct spread spectrum technology is a wideband modulation in which the original bit sequence is converted into a pseudo-random spreading sequence [18]. As a result, we obtained the transmission of a significantly larger number of extended (direct) sequence bits in the same time interval. Consequently, the frequency range for transmitting signals increases in proportion to the length of the spreading sequence [19].

### 3.1. Direct Spread Spectrum Technology in Telecommunications

The fundamental Shannon–Hartley theorem sets a constraint on the channel capacity, i.e., to the upper limit of the maximum amount of information that can be transmitted in a communication system with a given frequency band and power of additive white Gaussian noise [52]:(1)$$C=\Delta F\log_{2}\left(1+\frac{P_{S}}{P_{N}}\right),$$
where:C—channel throughput, bit/s;ΔF—frequency band, Hz;$P_{S}$—desired signal power;$P_{N}$—additive white Gaussian noise power;$\frac{P_{S}}{P_{N}}$—signal power to noise ratio (SNR).

From Equation (1), we see that at low SNRs, the throughput can be increased only by expanding the frequency band, and direct spread spectrum technology is well suited for this. Indeed, using very long spreading sequences, even at a very low SNR value, high-speed information transmission can be realized. This circumstance can also be used, e.g., to build environmentally friendliness of communication systems, i.e., when $P_{S}\approx P_{N}$. Then, using spreading sequences with a bit length $10^{8}$ or more, it is possible to realize information transfer at rates of tens and hundreds of Mb/s.

Let us explain the technology of spreading the spectrum with a direct sequence in the following simple way.

Consider a set $s=(s_{1},s_{2},\ldots ,s_{k})$ of pseudo-random bit sequences (vectors):$$\begin{array}{c} s_{1}=(s_{1,0},s_{1,1},\ldots ,s_{1,n-1}),\\ s_{2}=(s_{2,0},s_{2,1},\ldots ,s_{2,n-1}),\\ \ldots ,\\ s_{k}=(s_{k,0},s_{k,1},\ldots ,s_{k,n-1}), \end{array}$$ represented, for example, in polar form, i.e.,
$$\forall i,j:s_{i,j}=\{\begin{array}{c} 1,\\ -1. \end{array}$$

Vectors are designed to spread the spectrum of the original message, so they are called chip codes or chipping codes. Each value $s_{i,j}$ is called a chip or sequence element.

The vectors $s_{1},s_{2},\ldots ,s_{k}\in s$ are formed so that their cross-correlation is negligible, i.e.,
(2)$$\forall i\neq j:\rho (s_{i},s_{j})=\sum_{v=0}^{n-1}s_{i,v}s_{j,v}\approx 0$$

For example, if the vectors $s_{1},s_{2},\ldots ,s_{k}$ are taken equal to the rows (or columns) of the Hadamard matrix, we obtain orthogonal chip codes, i.e.,
$$\forall i\neq j:\rho (s_{i},s_{j})=0$$

There are other ways to generate chip codes, for example, when the elements $s_{i,j}$ take a random value over an interval [−1,1]. However, condition (2) for the generation of spreading sequences is decisive.

Suppose the message is made up of k bits, i.e., $m_{1},m_{2},\ldots ,m_{k}$, e.g., let us write the information bits $m_{i},i=1,2,\ldots ,k$ also in polar form, i.e.,
$$\forall i:m_{i}=\{\begin{array}{c} 1,\\ -1. \end{array}$$

Modulation of the information bits $m_{i},i=1,2,\ldots ,k$ can be achieved in different ways, e.g., with polar notation according to the expression:(3)$$\forall i:M_{i}=m_{i}s_{i}=(m_{i}s_{i,0},m_{i}s_{i,1},\ldots ,m_{i}s_{i,n-1})$$

In this way, instead of one binary element, $m_{i}$ a vector (sequence) $M_{i}$ of binary elements n, is transmitted.

Let us assume that the durations of an element $m_{i}$ and its extended chip version $M_{i}$ are the same. Suppose, for example, that they are equal to T seconds. Then the duration of one chip $s_{i,j}$ will be n times less: $T_{chip}=\frac{T}{n}$. Therefore, the chips are transmitted at a frequency
$$F_{chip}=\frac{1}{T_{chip}}=\frac{n}{T}\;\mathrm{Hertz}.$$

This frequency is n times greater than the frequency of information bits $F=\frac{1}{T}$. Thus, we obtained the spread of the frequency of the original message by n times.

The multiplying of the signal duration by its frequency is called the base B. If B>1, then the signals are called complex.

For our case, we have
$$B=T_{chip}F_{chip}=n>1$$
and direct spreading signals are complex signals.

Several information bits can be simultaneously transmitted to the communication channel, i.e., the receiver obtains an additive mixture:(4)$$Mix=N+\sum_{j=1}^{k}M_{j}$$
where the symbol N denotes the sequence of random elements caused by noise (for example, additive white Gaussian noise).

The receiver has synchronized copies of the chip codes $s_{1},s_{2},\ldots ,s_{k}\in s$. In order to restore the information bits, it is necessary to calculate the correlation between the received signal and the corresponding chip codes. For example, to restore the value $m_{i}$ in (3), the receiver computes $\rho (s_{i},Mix)$. Equation (2) is linear, i.e., write down
(5)$$\rho (s_{i},Mix)=\rho (s_{i},N)+\rho (s_{i},\sum_{j=1}^{k}M_{j})$$

For random noise, N, we have:(6)$$\rho (s_{i},N)\approx 0$$

Moreover, $\forall i\neq j:\rho (s_{i},s_{j})\approx 0$; therefore,
(7)$$\rho (s_{i},Mix)\approx \sum_{j}\rho (s_{i},m_{j}s_{j})\approx \rho (s_{i},m_{i}s_{i})$$

In this way, we can write the information bit recovery rule
(8)$$m_{i}=\rho (s_{i},Mix)=\{\begin{array}{c} -1,\;\rho (s_{i},Mix)<0;\\ +1,\;\rho (s_{i},Mix)>0. \end{array}$$

Note that in the above reasoning, individual bits $m_{i},i=1,2,\ldots ,k$ may belong to different senders. In this case, code division multiple access is implemented [18].

Let us note the following advantages of using direct spread spectrum technology:Resistance to unintended or intended jamming;Sharing a single channel among multiple users;Reduced signal/background-noise level hampers interception;Determination of relative timing between transmitter and receiver and much more.

We are considering the use of this technology in steganography, i.e., to hide informational messages in multimedia files.

### 3.2. Direct Spread Spectrum in Steganography

Direct spread spectrum technology has long been used in steganography.

In the first works [20,21,40], a general scheme of a steganosystem with the direct spreading of the spectrum was proposed. The authors used cover images, but all their reasoning can be extended to other multimedia data.

The general idea of such a stegosystem is as follows (Figure 1).

In order to hide the data, the following steps are performed:

Step 1The information message is presented in polar form $m_{i}$;Step 2$m_{i}$ multiplied bit by bit by the spreading sequence as in (3);Step 3Multimedia data (images, audio, video, etc.) are interpreted as noise in the communication channel. Using our notation, this is N in (4).Step 4Further, these data are hidden in the cover data according to rule (4). However, we currently denote:
N—multimedia data (container, cover), inside which the message is hidden;Mix—modified multimedia data (filled container) after hiding messages in them.

In order to restore the data, the following steps are performed:
Step 1Correlation is calculated.
-The product is calculated $s_{i}Mix$;-The sum is calculated $\rho (s_{i},Mix)=\sum_{v=0}^{n-1}s_{i,v}Mix_{v}$;
Step 2If Equations (6) and (7) are true, then the value of the recovered bit is calculated by the Equation (8).Step 3The recovered message is presented in a custom view.

It should be noted that even in the first works, the authors encountered significant difficulties. For example, in [53] was shown that the error rate in the extracted messages is very high (see, for example, Table 2 on page 12 of [53]). BER can be reduced in different ways, for example, by using filtering, correction codes, etc. The simplest is to increase the power of chip codes. For this, Equation (4) can be rewritten as
(9)$$Mix=N+P\sum_{j=1}^{k}M_{j}=N+\sum_{j=1}^{k}m_{j}(Ps_{j})$$
where the value P specifies the power amplification factor of the chip codes $s_{j}=(s_{j,0},s_{j,1},\ldots ,s_{j,n-1})$.

By increasing the power, P, it is actually possible to achieve a decrease in BER. However, this inevitably leads to distortion of the container N. For example, for k=10 and P=10, each value of the cover file N may be skewed by ±kP=100. This may present a real problem because BER is still high. For example, in [53], it is shown that even with P=100 an error rate is more than 10% (see Table 2 from [53]).

We carried out detailed studies of this problem in [23,24]. In particular, we repeated the experiments with images from [21,40,53,54]. We studied the effect of different ways of forming spreading sequences on the BER value. The general conclusions about the high BER were confirmed, and the explanation for this phenomenon is, in our opinion, the following. The first term in (4) presents a certain implementation of random additive white noise N. This is indeed the case for spread spectrum communications systems. Those assumption (6) is fulfilled, which gives an almost error-free recovery of the message according to rule (8). However, in the stagnosystem, in (4), multimedia data are meant. This is usually highly redundant, highly correlated data. This is not a random realization of Gaussian noise, and assumption (6) is often not met. Obviously, this leads to errors in the recovered bits according to rule (8). In works [55,56], an effective way to combat this phenomenon was proposed. We proposed to form chip sequences in a special, adaptive way. Indeed, if we take into account the statistical properties of multimedia data (in [55,56], we used cover images), then it is possible to form chip codes $s_{1},s_{2},\ldots ,s_{k}$, in such a way that condition (6) is guaranteed to be satisfied. In this case, it is possible to ensure an almost error-free extraction of information messages according to rule (8).

In this article, we continued our experiments. We used audio covers and experimented with different ways to generate chip codes. We showed that the variety of ways of generating spreading sequences produces different BER values. This is an area for possible optimization of the stegosystem, and we showed which chip codes are really best used to hide information messages in audio containers.

### 3.3. Initial Data for Hiding Messages

We used various audio carriers and various families of chip codes as the initial data for our experiments. We were guided by the works [21,40] and our papers [23,24] when choosing the initial data and research methodology.

An audio signal recorded in the digital WAV format [10,11] was used as an audio container. In our experiments, we used an audio file with a student anthem (Gaudeamus Igitur) downloaded from an internet resource [57] and converted to WAV format.

WAV audio is most commonly encoded using linear pulse code modulation (LPCM). Such a cover file contains uncompressed audio signal data; its main characteristics are:

the number of channels (streams) of audio data $N_{channel\;}$. We used stereo signals, i.e.,$N_{channel\;}=2$;sampling frequency $f_{d}$. Our audio signals from $f_{d}=22050$;the number of bits allocated to encode each discrete sample of the audio signal. We used covers from B=8 bit.

As input information messages, we used plain text files (steganography textbooks), processed and presented in binary polar form.

In order to hide information, various spreading sequences were used, studied by many authors, for example, in [21,40] and also in our previous works [23,24]. Let us consider these chip codes in more detail.

### 3.4. Used Chip Codes

In this article, various methods for generating chip codes were considered, and their influence on the characteristics of a steganosystem was studied. Specifically, we explored five different ways to generate spreading sequences.

Spread sequences from [21,40]

In the basic works [21,40] on Spread Spectrum Steganography, a nonlinear rule for the formation of chip codes $s=(s_{1},s_{2},\ldots ,s_{k})$ was used. In particular, each chip $s_{i,j}$ was formed as a realization of a random variable distributed according to the standard normal law.

For this, the rule was proposed:(10)$$s_{i,j}=\{\begin{array}{c} \Phi^{-1}(u_{i,j}),\;m_{i}=-1;\\ \Phi^{-1}(u\prime_{i,j}),\;m_{i}=1, \end{array}$$
where
$$u\prime_{i,j}=\{\begin{array}{c} u_{i,j}+0.5,\;u_{i}<0.5;\\ u_{i,j}-0.5,\;u_{i}\geq 0.5, \end{array}$$

$u_{i,j}$—implementation of a random variable uniformly distributed over the interval [0,1];$\Phi^{-1}$—inverse cumulative distribution function for a standard Gaussian random variable.

In our experiments, we used chip codes of length n=1024 generated according to rule (10).

Spreading sequences with a Gaussian chip distribution (10) were also studied by us earlier in relation to cover images in [23,24]. We are expanding our research to the case of audio containers now.

2.Chip codes from random numbers uniformly distributed over the interval [−1, 1]

We also implemented the formation of spreading sequences of length n=1024 from random numbers uniformly distributed over the interval [−1, 1]. Each chip $s_{i,j}$ is formed randomly with equal probability and independently of other values.

3.Chip codes from normally distributed random numbers

This method of forming spreading sequences of length n=1024 consists in generating normally distributed random numbers. In fact, the second and third methods are very similar. Each chip $s_{i,j}$ is randomly formed. The second method uses a uniform distribution on the interval [−1, 1], and the third one uses a standard normal distribution. The third method is also similar to the first, but the rule for generating chip codes is somewhat different.

4.Binary chip codes generated by a pseudo-random bit generator

The simplest, in our opinion, is the formation of individual chips using a pseudo-random bit generator. We converted the generated bits to polar form and formed spreading sequences of length n=1024.

5.Walsh–Hadamard spreading sequences

These are discrete sequences of length $n=2^{w}$, w=1,2,…, which can be formed as rows (or columns) of the Hadamard matrix $H_{n}$ of order $n=2^{w}$:(11)$$H_{2^{w}}=\left[\begin{array}{cc} H_{2^{w-1}} & H_{2^{w-1}}\\ H_{2^{w-1}} & -H_{2^{w-1}} \end{array}\right],\;H_{1}=\left[1\right].\;$$

For example, for n=4, we have four chip codes:$$s_{1}=(1,1,1,1),\;s_{2}=(1,-1,1,-1),\;s_{3}=(1,1,-1,-1),\;s_{4}=(1,-1,-1,1).$$

In our experiments, we used spreading sequences with a length of $n=2^{10}=1024$. Therefore, we recursively generated $H_{1024}$ according to rule (11) and formed $s=(s_{1},s_{2},\ldots ,s_{k=1024})$ as a set of strings $H_{1024}$.

The Walsh–Hadamard sequences form an orthogonal system, i.e.,
$$\forall i\neq j:\rho (s_{i},s_{j})=0$$
which is in good agreement with (2).

Nevertheless, the main disadvantage of such chip codes should be pointed out. The number of different orthogonal sequences of length n cannot exceed n. Therefore, to hide a large number of bits in the same container, this generation method may not be sufficient.

### 3.5. Indicators of Effectiveness of Hiding Messages

In order to evaluate the efficiency of the steganosystem, it is necessary to evaluate the distortions in the recovered messages, as well as the distortions of the original container. Therefore, we experimentally estimated the bit error rate (BER) in recovered messages. We also estimated audio cover distortions by mean squared error (MSE) and peak signal-to-noise ratio (PSNR). We did not use other indicators of distortion of the audio cover (for example, percentage residual deviation, autocorrelation analysis of stegodata with original data, etc.), as this can be obtained after processing our results.

These are the most common metrics used in many related works.

Bit Error Rate, BER

The bit error rate is calculated using the equation:(12)$$BER=\frac{k_{error}}{k}$$
where $k_{error}$ is the number of erroneously recovered bits, k is the total number of bits in the information message.

2.Mean squared error, MSE

In order to estimate the container distortion, the root mean square error (MSE) is used, i.e., the averaged square of the difference between the original multimedia data and the cover data obtained after the message was hidden. For an audio signal represented by a set of n discrete samples, *MSE* is calculated by the equation:


(13)
$$MSE=\frac{1}{n}\sum_{i=0}^{n-1}{\left[Mix_{i}-N_{i}\right]}^{2}$$


3.Peak signal-to-noise ratio, PSNR

A more visual characteristic of cover distortion is the ratio of the maximum possible signal power to the power of the distorting noise. Usually, this characteristic is expressed on a logarithmic scale and calculated by the equation:(14)$$\begin{array}{l} PSNR=10\cdot \log_{10}\left(\frac{N_{\max}^{2}}{MSE}\right)=20\cdot \log_{10}\left(\frac{N_{\max}}{\sqrt{MSE}}\right)=\\ =20\cdot \log_{10}\left(N_{\max}\right)-10\cdot \log_{10}\left(MSE\right), \end{array}$$
where $N_{\max}$ is the maximum possible value of the signal N.

In our case, by N we mean multimedia data, i.e., an audio signal represented by a set of discrete samples. If each discrete sample is coded in B bits, then (in linear pulse code modulation, PCM) the maximum possible value of $N_{\max}=2^{B}-1$. In our experiments, we used the simplest audio signals with B=8, i.e., $N_{\max}=255$. Equation (14) for this value takes the form:$$PSNR=20\cdot \log_{10}\left(255\right)-10\cdot \log_{10}\left(MSE\right).$$

## 4. Results

### 4.1. Experimental Evaluation of Hidden Message Distortion

In Figure 2, Figure 3, Figure 4, Figure 5 and Figure 6, the results of our experiments on BER estimation are shown for different methods of generating chip codes of length n=1024. We changed the number k of hidden information bits and the coefficient P for amplifying the power of the spreading signals in (9).

The obtained results of experimental studies correspond to the points in the given diagrams. Each point corresponds to a value averaged over 5000 tests. For better perception, these and all subsequent figures also show trend lines.

The analysis of the results obtained (see Figure 2, Figure 3, Figure 4, Figure 5 and Figure 6) enables us to make the following observations.

First, an increase in the coefficient P makes it possible to achieve low BER values for any of the considered classes of chip codes. At the same time, chip codes with generation rules №1, 2 and 3, 4 give similar BER values, i.e., for this indicator, such spreading sequences are equivalent to each other.

Walsh–Hadamard sequences give significantly better results in terms of error rates. Even at low values of P, these chip codes provide low BER.

The second observation is that the BER increases as the number of bits in the hidden message increases. However, different chip codes give different results. For example, spreading sequences №1 and №2 at k=10 provide BER<10% only at P>8. For chip codes №3 and №4, the same is provided with a lower power gain, for example, for P>4. The Walsh–Hadamard sequences even with P=1 and k=20 provide BER<1%.

In order to compare the BER dependences corresponding to different methods of forming chip codes, Figure 7 shows diagrams BER(P). These are averaged over 5000 test results for k=20 bits.

The analysis of the results shown in Figure 7 demonstrates that the techniques for generating chip codes №1 and №2, as well as №3 and №4, do indeed give similar BER values. However, such a difference in the BER value is observed only for large k values. With a small number of hidden bits, the BER values for chip codes №1–4 are practically the same. This is clearly seen in Figure 8. Thus, for small values of k, it is really possible to achieve a low error rate for practically any method of forming chip codes.

Walsh–Hadamard sequences show significantly better BER results. For this family of chip codes, the error rate, other things being equal, is one to two orders of magnitude lower. At the same time, BER can be significantly reduced only for large P values, and this will inevitably lead to container distortion.

### 4.2. Experimental Evaluation of Container Distortion

We used MSE and PSNR to assess container distortion. The obtained experimental results indicate that using chip codes generated according to rules 1, 3, 4, and 5 leads to approximately equal distortions of containers. The corresponding dependences of MSE(k) and PSNR(k) for different values are shown in Figure 9 and Figure 10.

Somewhat less distortion of the container is provided by using the rule for generating chip codes №2; the corresponding dependencies are shown in Figure 11 and Figure 12.

The analysis of Figure 9, Figure 10, Figure 11 and Figure 12 enables us to make the following observations.

First, an increase in the number of hidden information bits leads to an inevitable increase in container distortion. Usually, PSNR values of about 30–40 dB are considered acceptable quality. If you focus on these values, then you can effectively hide no more than 10–20 information bits.

Secondly, an increase in the power gain P also leads to an increase in container distortion. If we focus on PSNR of about 30.40 dB, then the values of P<8 are acceptable (with k<4).

Thus, the obtained results of experimental studies show that by increasing the length k of the hidden message, we inevitably distort the container. Raising P in order to reduce BER also leads to an increase in container distortion. Thus, there are several conflicting factors that directly affect the efficiency of the steganosystem. In order to find a compromise, it is necessary to study the influence of these factors on each other.

### 4.3. Correlation of Error Rate and Container Distortion, Finding a Compromise

In order to find a compromise solution, let us study the relationship between BER and PSNR.

In Figure 13, Figure 14, Figure 15, Figure 16 and Figure 17 the dependences obtained as a result of averaging over 5000 tests are shown.

## 5. Discussion

The analysis of the data shown in Figure 2, Figure 3, Figure 4, Figure 5, Figure 6, Figure 7, Figure 8, Figure 9, Figure 10, Figure 11, Figure 12, Figure 13, Figure 14, Figure 15, Figure 16 and Figure 17 results enables us to draw the following conclusions.

First, it is difficult to implement information hiding without significant container distortion. The above dependences of BER(PSNR) clearly demonstrate this. By increasing the volume of the hidden message k and/or the power amplification factor P, we certainly distort the container, i.e., decrease PSNR. For example, chip codes №1 and №2 allow hiding only 1–2 bits of the message when PSNR>30 dB and BER<10%. Spreading sequences №3 and №4 are somewhat better; they make it possible to hide 1–2 message bits when PSNR>35 dB and BER<10%. However, this is still not enough. Finally, the Walsh–Hadamard signals perform well. If they are used, it is possible to hide 16 or more message bits and provide PSNR>30 dB and BER<1%. This confirms the earlier conclusion about the preference of such chip codes for hiding information. This is also consistent with our findings from Spread Spectrum Image Steganography [23,24].

The second conclusion is a possible trade-off between the amount of hidden data and the introduced distortion of the audio container. For example, as follows from Figure 17, halving the k value leads to an increase in PSNR by about 5 dB. This can be used to configure the desired stegosystem parameters in a specific practical application.

The third and probably the most important conclusion is that it is almost impossible to achieve high volumes of hidden information with the known techniques. For example, even for the best case shown in Figure 17, with PSNR=35 dB and BER<1%, only k≤16 bits of information can be hidden. It is possible to improve these values by traditional techniques only by increasing the length of the chip codes, which inevitably leads to an increase in the computational complexity of the transformations. This could, however, become a problem. Indeed, to restore each closed beta according to rule (8), it is necessary to calculate the correlation according to Equation (5). Therefore, you need to perform n additions and n multiplications, where n is the length of the chip codes. Our experiments correspond to the case of n=1024, and we had a performance comparable to other modern data hiding techniques. However, as n increases, performance decreases, which can significantly complicate practical implementation.

Another way is the implementation of adaptive techniques, for example, as in [55,56], or the use of new, advanced methods of hiding information, for example, based on addressing chip codes [58]. These and other methods are a promising direction for our further research.

## 6. Conclusions

In the present paper, we explored techniques for hiding information in audio containers using a direct spread spectrum. We considered various spreading sequences (chip codes) and studied their influence on the error rate in recovered messages. We also investigated the distortion of the audio container in terms of MSE and PSNR.

The results clearly demonstrate the promising characteristics of this direction. Precisely, in each segment of the container, we managed to hide a part of the information message reliably. At the same time, the BER, MSE, and PSNR values are within an acceptable range. The best characteristics were shown by Walsh–Hadamard chip codes, which showed the lowest error rate (with comparable container distortions).

At the same time, it should be noted that there exist objectively conflicting factors that negatively affect one another. For example, by increasing the volume of hidden messages, we inevitably increase the BER and distort the container. By increasing the power of the chip codes, it is possible to reduce the BER. However, this distorts the container even more. This is consistent with known results in steganography based on direct spectrum expansion. For example, in [41,53], it is shown that by increasing the power of chip codes, it is really possible to reduce BER, but the distortion of the covers becomes very significant. We managed to present the availability of compromise solutions. This is especially true for Walsh–Hadamard chip codes, for which the range of possible solutions is much wider than for other spreading sequences. Nevertheless, it should be noted that when using traditional data hiding techniques, the volume of hidden messages cannot be large. The search for new ways of hiding information is promising, e.g., based on the adaptive formation of spreading sequences [56], advanced techniques for addressing chip codes [58], etc.

## Figures and Tables

**Figure 1 sensors-22-03115-f001:**
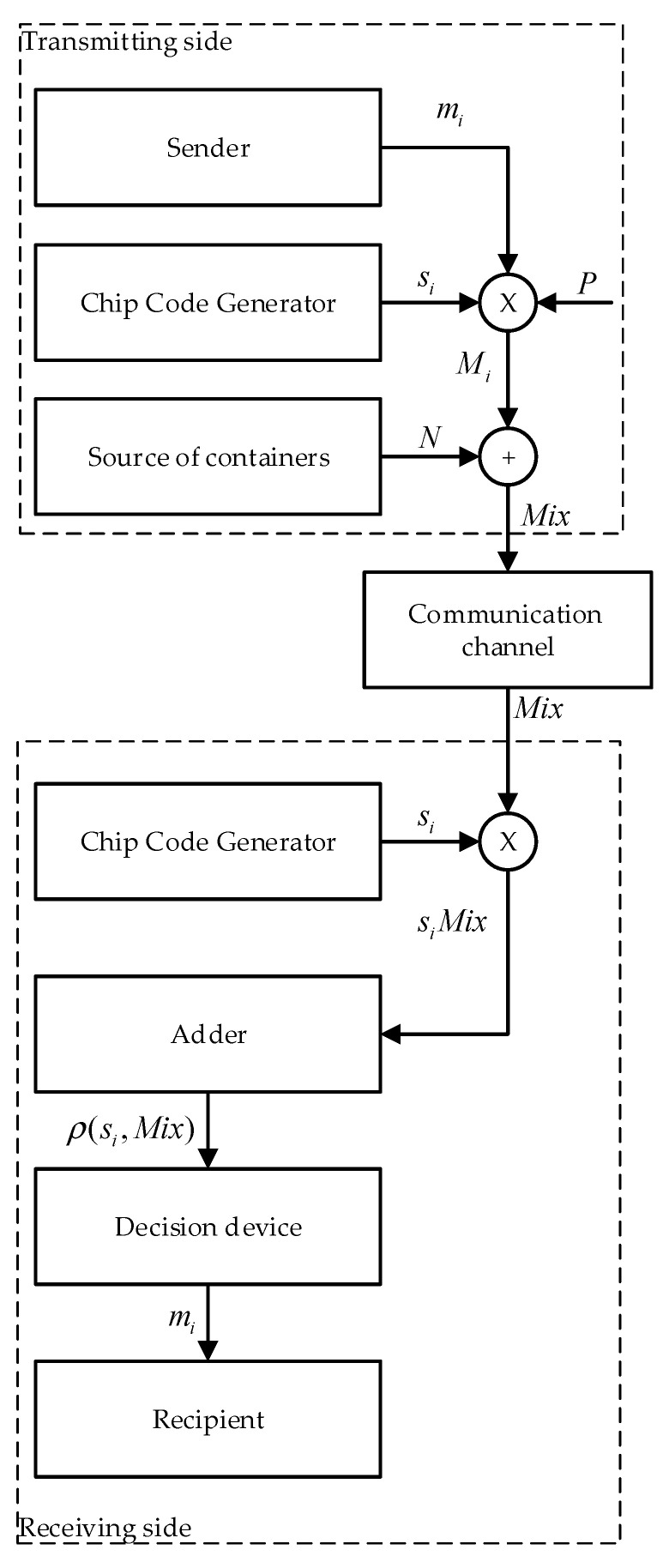
Block diagram of concealment and recovery.

**Figure 2 sensors-22-03115-f002:**
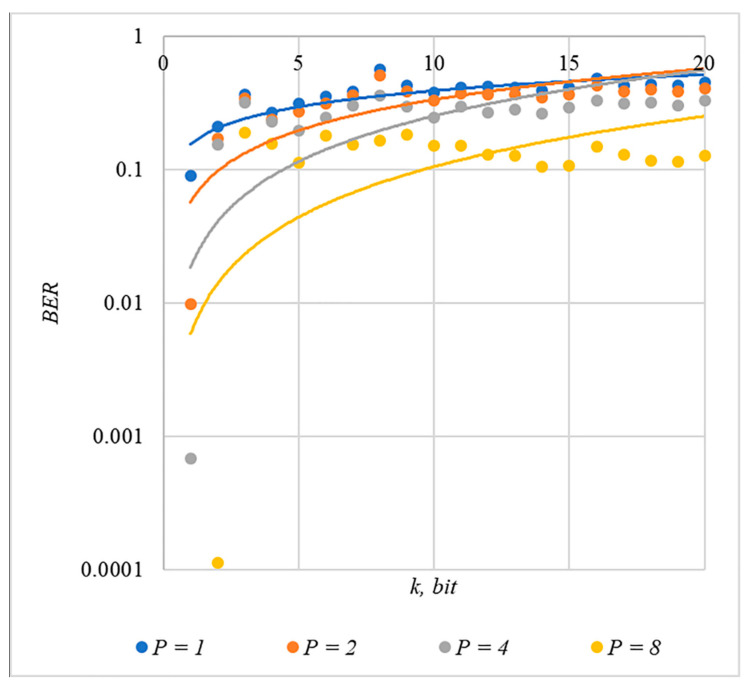
*BER*(*k*) dependencies for different values of P rule for generating chip codes №1.

**Figure 3 sensors-22-03115-f003:**
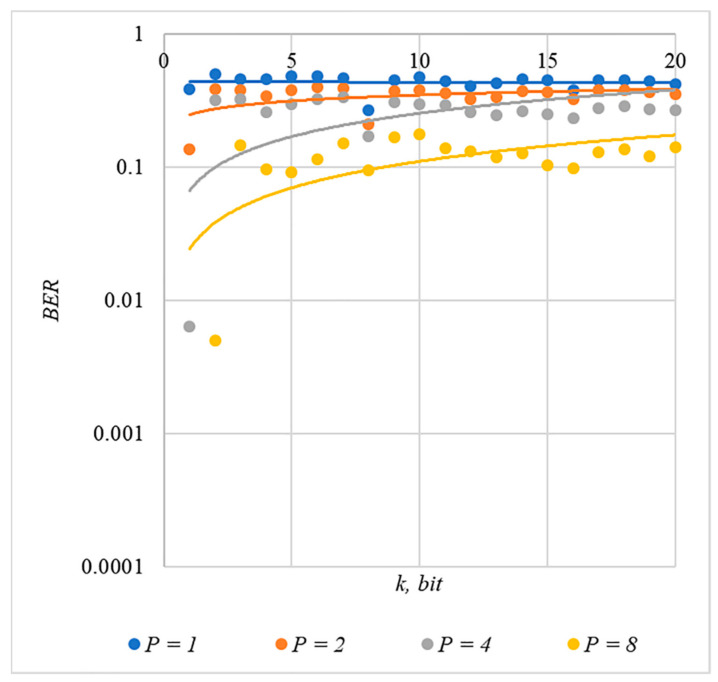
*BER*(*k*) dependencies for different values of P rule for generating chip codes №2.

**Figure 4 sensors-22-03115-f004:**
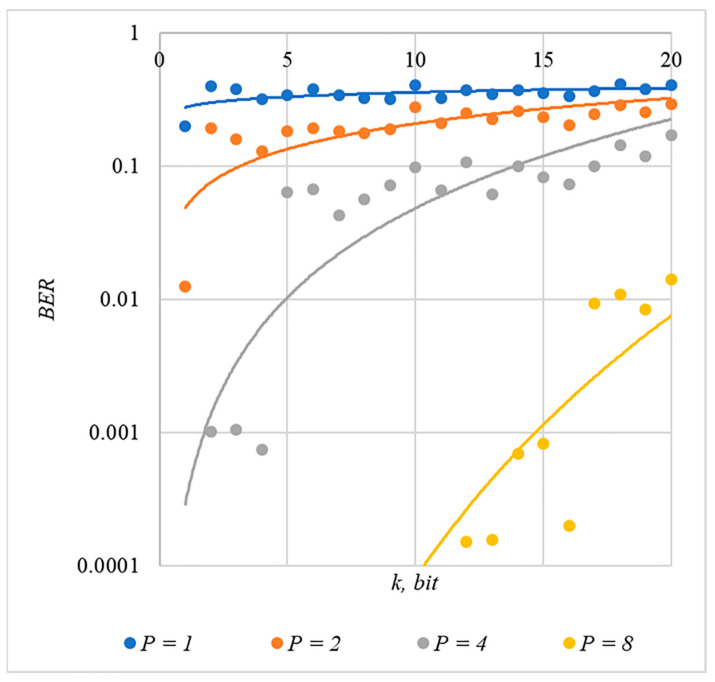
*BER*(*k*) dependencies for different values of P rule for generating chip codes №3.

**Figure 5 sensors-22-03115-f005:**
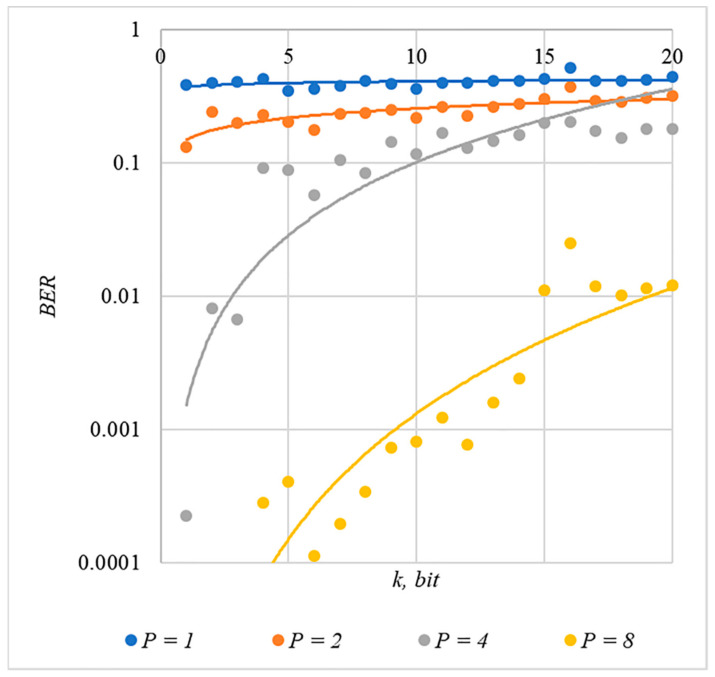
*BER*(*k*) dependencies for different values of P rule for generating chip codes №4.

**Figure 6 sensors-22-03115-f006:**
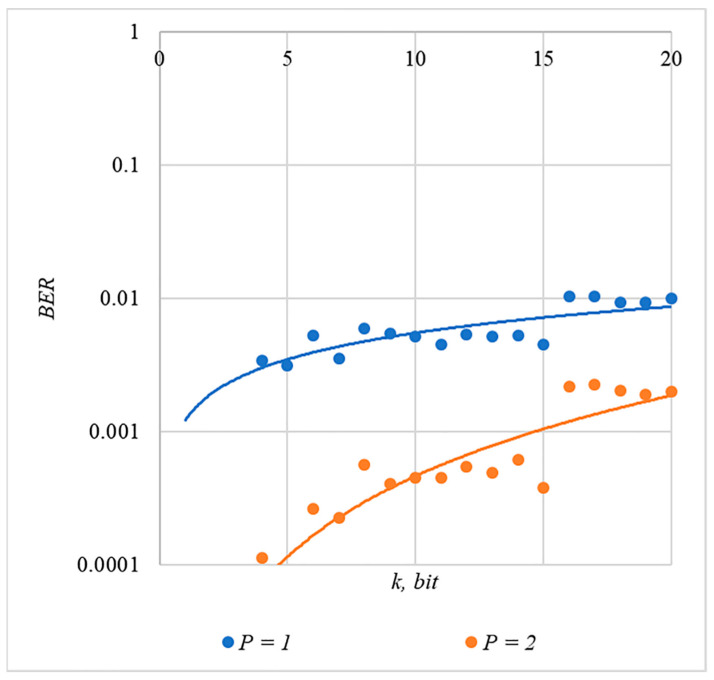
*BER*(*k*) dependencies for different values of P rule for generating chip codes №5.

**Figure 7 sensors-22-03115-f007:**
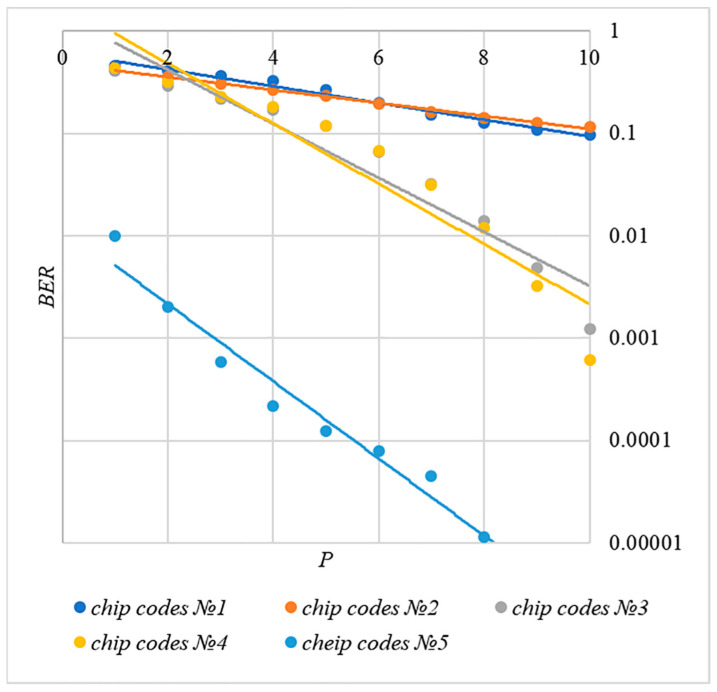
*BER*(*P*) dependencies for different rules for generating chip codes, k=20.

**Figure 8 sensors-22-03115-f008:**
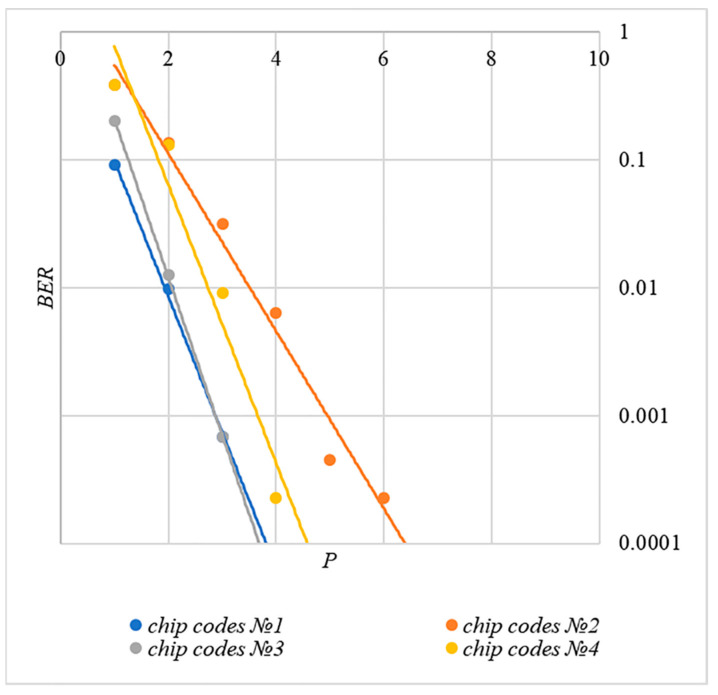
*BER*(*P*) dependencies for different rules for generating chip codes, k=1.

**Figure 9 sensors-22-03115-f009:**
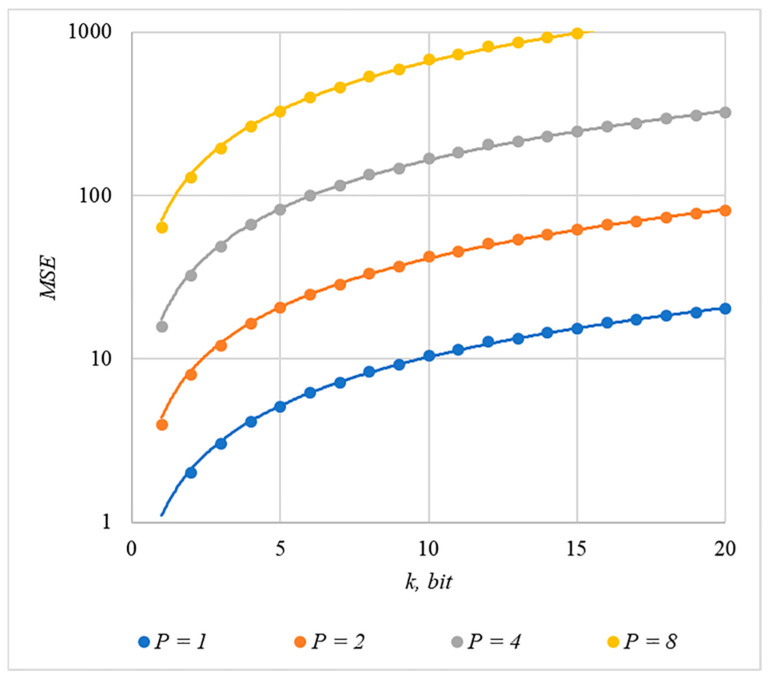
*MSE*(*k*) dependencies for different values of P rules for generating chip codes №1, 3, 4, and 5.

**Figure 10 sensors-22-03115-f010:**
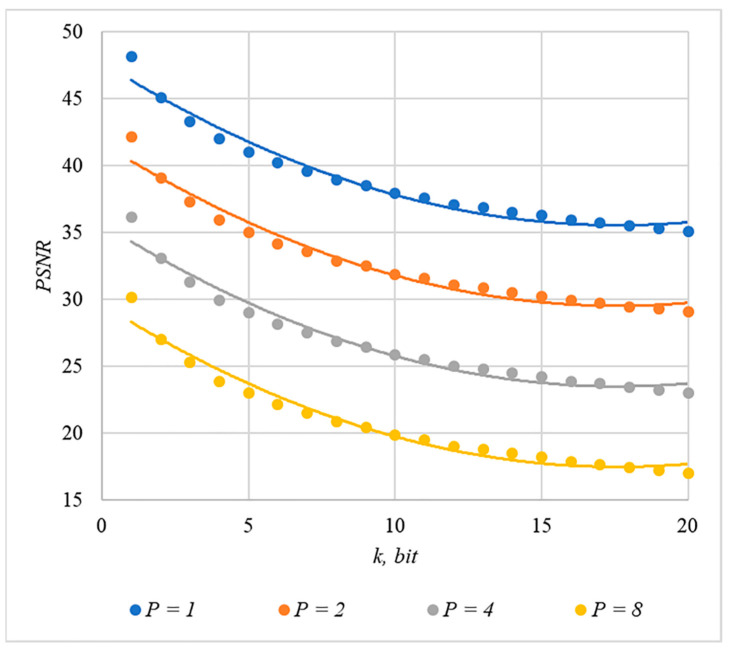
*PSNR*(*k*) dependencies for different values of P rules for generating chip codes №1, 3, 4, and 5.

**Figure 11 sensors-22-03115-f011:**
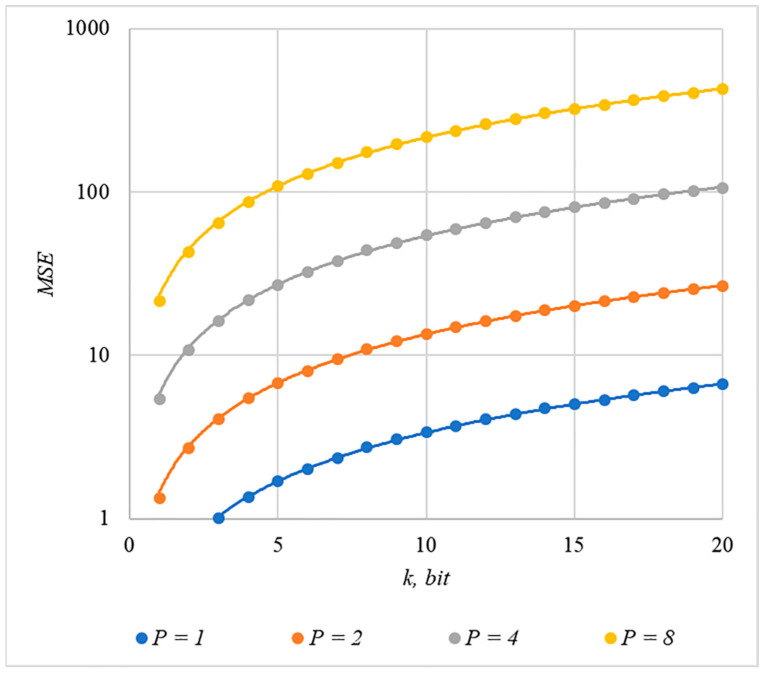
*MSE*(*k*) dependencies for different values of P rule for generating chip codes №2.

**Figure 12 sensors-22-03115-f012:**
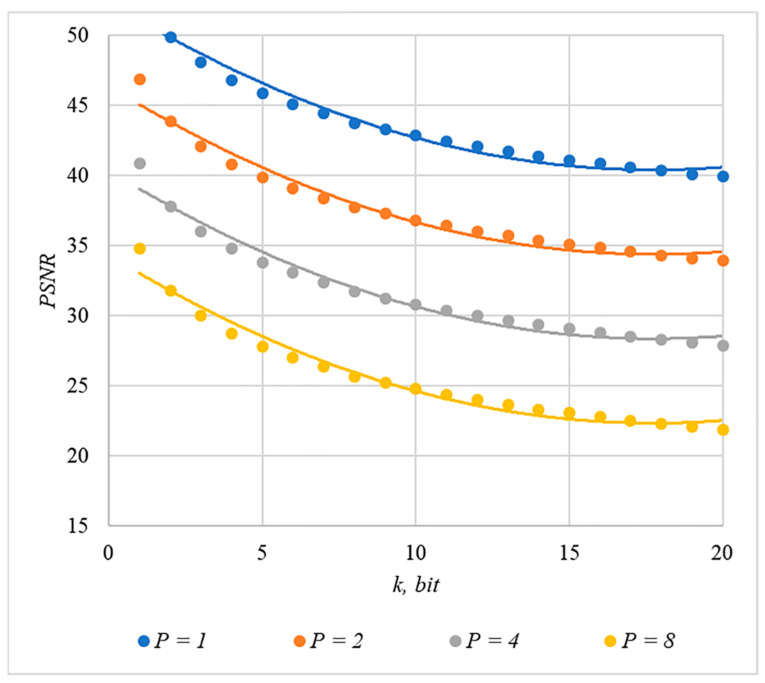
*PSNR*(*k*) dependencies for different values of P, rule for generating chip codes №2.

**Figure 13 sensors-22-03115-f013:**
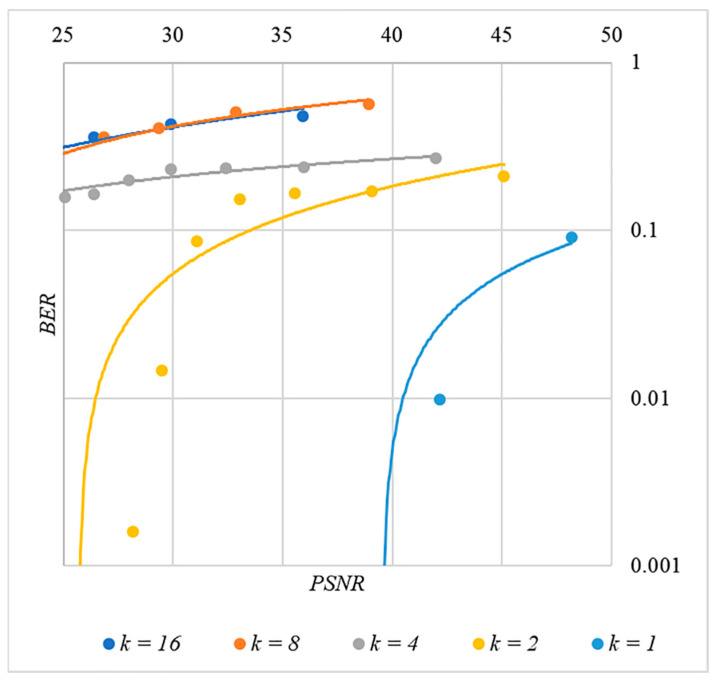
*BER*(*PSNR*) dependencies for different values k rule for generating chip codes №1.

**Figure 14 sensors-22-03115-f014:**
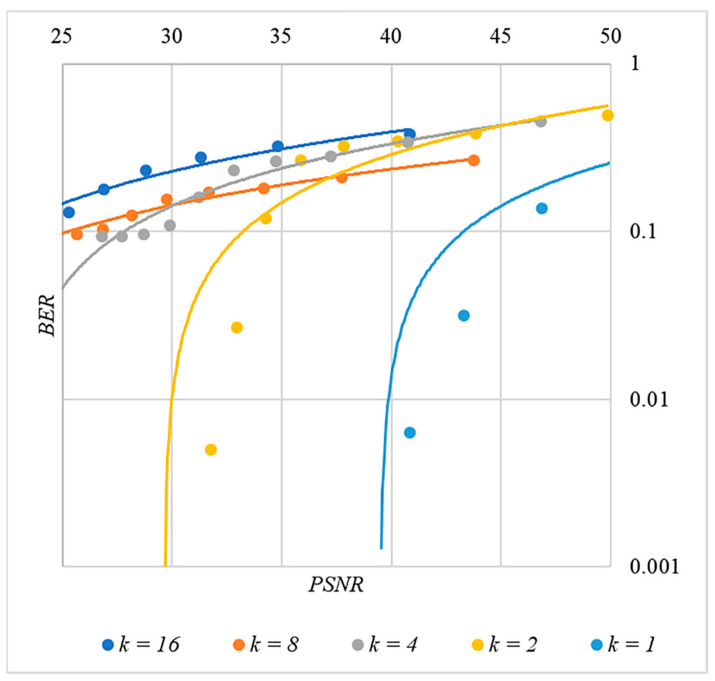
*BER*(*PSNR*) dependencies for different values k rule for generating chip codes №2.

**Figure 15 sensors-22-03115-f015:**
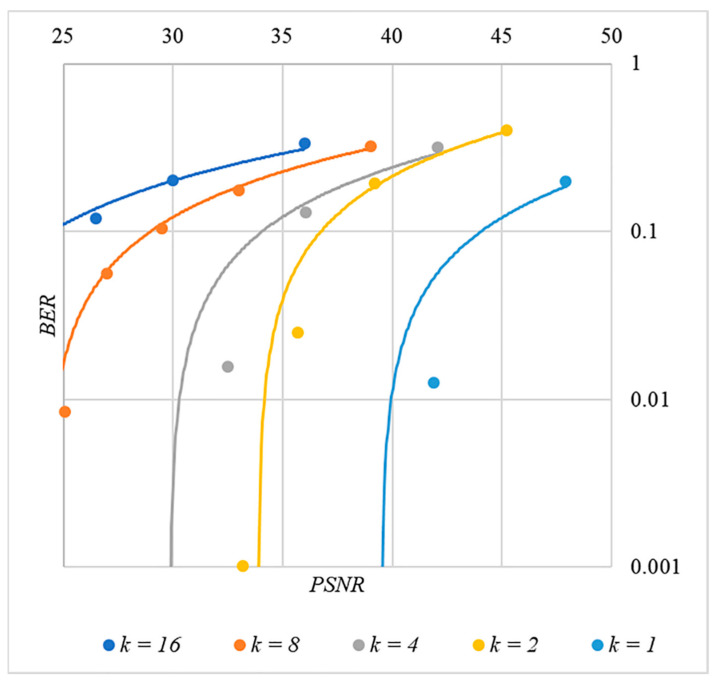
*BER*(*PSNR*) dependencies for different values k rule for generating chip codes №3.

**Figure 16 sensors-22-03115-f016:**
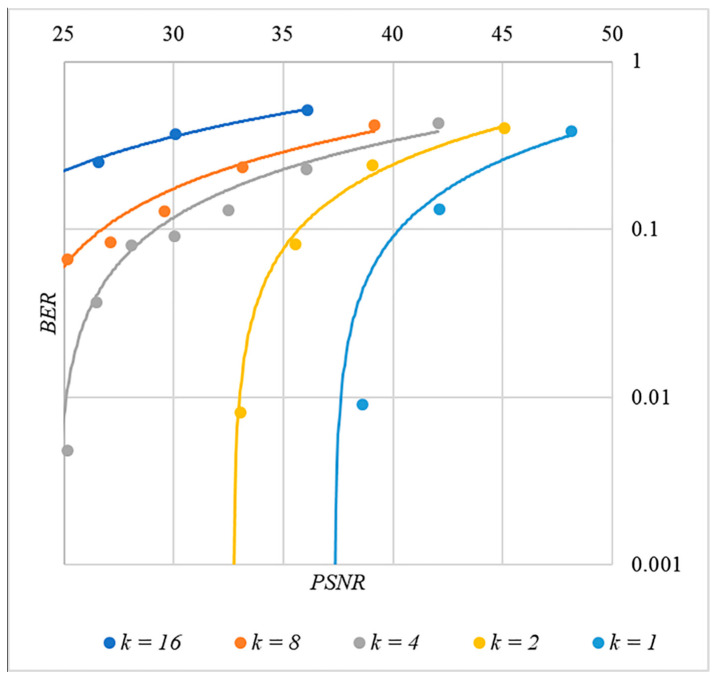
*BER*(*PSNR*) dependencies for different values k rule for generating chip codes №4.

**Figure 17 sensors-22-03115-f017:**
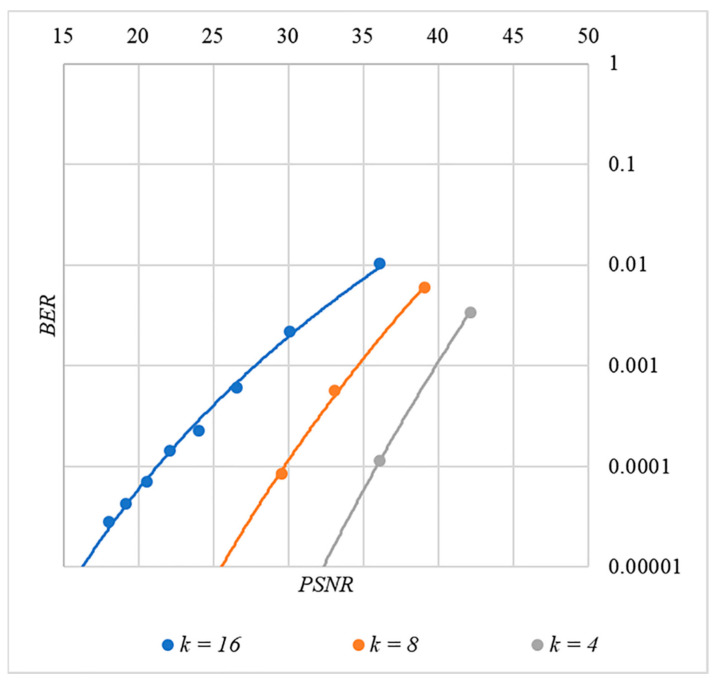
*BER*(*PSNR*) dependencies for different values k rule for generating chip codes №5.

## Data Availability

Not applicable.

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
