# Peer review of "Direct Spread Spectrum Technology for Data Hiding in Audio†"

_sensors, 2022, doi:10.3390/s22093115_

Round 1

Reviewer 1 Report

The direct spread spectrum is applied to the data hiding for audio containers. The presentation is clear with fine simulations. I recommend the acceptance of this paper after revision.

  1. More introduction related to the data hiding based on the audio process may broaden the scope of this paper such as

         1.1 Masashi Unoki, Kuniaki Imabeppu, Daiki Hamada, Atsushi Haniu, and Ryota Miyauchi, “Embedding Limitations with Digital-audio Watermarking Method Based on Cochlear Delay Characteristics”, Journal of Information Hiding and Multimedia Signal Processing, Vol. 2, No. 1, pp. 1-23, January 2011

         1.2 Mun Chun Lee and Chee Yong Lau, “Three Orders Mixture Algorithm of Audio Steganography Combining Cryptography”, Journal of Information Hiding and Multimedia Signal Processing, Vol. 9, No. 4, pp. 959-969, July 2018

          1.3 Hussein A. Nassrullah, Wameedh N. Flayyih and Mohammed A. Nasrullah, “Enhancement of LSB Audio Steganography Based on Carrier and Message Characteristics”, Journal of Information Hiding and Multimedia Signal Processing, Vol. 11, No. 3, pp. 126-137, September 2020 

Author Response

We appreciate the reviewer's high assessment of our work. We have expanded the literature review and indicated recommended sources.

Reviewer 2 Report

Introduction is not motivating the interested readers. Develop it further.

Inadequate related survey is done, develop it further and discuss some recent related works.

The authors need to present their proposed information hiding method through a flowchart and in proper step-wise form.

Percent Residual Deviation, & Auto-correlation analysis of stego data with original data is to be computed, do comparison also.

Performance comparison with some state of the art techniques is required.

Author Response

Introduction is not motivating the interested readers. Develop it further.

We have expanded and refined the introduction. We have added motivation to our research and indicated its relevance.

Inadequate related survey is done, develop it further and discuss some recent related works.

We have expanded the literature review and reviewed recent related works.

The authors need to present their proposed information hiding method through a flowchart and in proper step-wise form.

We have added a block chart and a step-by-step form for hiding and restoring data.

Percent Residual Deviation, & Auto-correlation analysis of stego data with original data is to be computed, do comparison also.

To assess the distortion of the covers, we used the MSE and PSNR indicators. These are the most common indicators used in many related works. Other indicators can be obtained after processing our results. We indicated this in the article.

Performance comparison with some state of the art techniques is required.

In section 5 "Discussion" we have added comments regarding the performance of our methods.

Reviewer 3 Report

the manuscript discussed using direct spread spectrum techniques for hiding information in audio. it is an interesting topic with decent related work previously done by the authors. 

I have a couple of comments:

  • using Related Work: please consider elaborating more on the existing techniques to shed some light on how different they are from your proposed method.
  • line 185: The paragraph about the DSS technology is kind of interrupting the flow of the ideas in the text. Please consider moving this to either the beginning of the section (3.1) or add it to the introduction(maybe after line 52)
  • I think that sections 3.3 & 3.5 belong to the results section more than the methods.
  • line 399 & 401: I think you need to add "&" between the figure numbers in Fig. 8.9 and Fig. 10.11
  • 4.12: "the following observations. " a : is expected

Author Response

Reviewer

Authors’ response

the manuscript discussed using direct spread spectrum techniques for hiding information in audio. it is an interesting topic with decent related work previously done by the authors. 

I have a couple of comments:

We appreciate the reviewer's high assessment of our work.

·       using Related Work: please consider elaborating more on the existing techniques to shed some light on how different they are from your proposed method.

·        

We have expanded the literature review and reviewed recent related works.

·       line 185: The paragraph about the DSS technology is kind of interrupting the flow of the ideas in the text. Please consider moving this to either the beginning of the section (3.1) or add it to the introduction(maybe after line 52)

We have inserted corrections according to the reviewer’s opinion.

·       I think that sections 3.3 & 3.5 belong to the results section more than the methods.

It may be so. However, for the flow of the material it is still more logical to preserve the existing consequence.

line 399 & 401: I think you need to add "&" between the figure numbers in Fig. 8.9 and Fig. 10.11

Corrections have been inserted.

·4.12: "the following observations. " a : is expected

Corrections have been inserted.